# In the Arms of Morpheus without Morphia; Mitigating the United States Opioid Epidemic by Decreasing the Surgical Use of Opioids

**DOI:** 10.3390/jcm10071472

**Published:** 2021-04-02

**Authors:** Karen Boretsky, Keira Mason

**Affiliations:** Department of Anesthesiology, Critical Care and Pain Medicine, Boston Children’s Hospital, 300 Longwood Avenue, Boston, MA 02115, USA; keira.mason@childrens.harvard.edu

**Keywords:** anesthesia, surgery, opioid epidemic, analgesia, regional anesthesia

## Abstract

The opioid epidemic is a major public health issue in the United States. Exposure of opioid naïve-patients to opioids in the perioperative period is a well-documented source of continued use with one in 20 opioid-naïve surgical patients continuing to use opioids beyond 90 days. There is no association with magnitude of surgery, major versus minor, and the strongest predictor of continued use is surgical exposure. Causal factors include over reliance on opioids for intraoperative and postoperative analgesia and excessive ambulatory opioid prescribing. Opioid-induced hyperalgesia can paradoxically result from intraoperative (anesthesia controlled) opioid administration. Increasing size of initial prescription is a strong predictor of continued use necessitating procedure specific supplies limited to under 3-days. Alternative multimodal pain management (non-opioid medications and regional anesthesia) that limit opioid use must be a high priority with opioids reserved for severe breakthrough pain. Barriers to implementation of opioid-sparing pathways include reluctance to adopt protocols and apprehension about opioid elimination. Considering the number of surgeries performed annually in the United States, perioperative physicians must aggressively address modifiable factors in surgical patients. Patient care pathways need to be constructed collaboratively by surgeons and anesthesiologists with continuing feedback to optimize patient outcomes including iatrogenic opioid dependence.

## 1. In the Arms of Morpheus without Morphia; Mitigating the United States Opioid Epidemic by Decreasing the Surgical Use of Opioids

Morpheus is the Greek god of dreams and when in his embrace, an individual enjoys a deep, peaceful sleep; referred to as “being in the arms of morpheus”. Morpheus scattered the seeds of poppies over weary mortals. Morphia or morphine is thus named after him. Hence begins the story of opioids. The history of the opioid epidemic in the United States is well documented and, in spite of the notoriety, the problem continues to escalate [1,2,3,4]. For the past 20 years, the United States has experienced a growing crisis of opioid abuse and addiction with a significant number of opioid overdose deaths attributed to a large increase in opioid prescribing for pain [5]. This excessive use was driven by several factors including the publication of journal articles touting opioids used for pain management as non-addictive, the focus by government regulatory agencies on better recognition and treatment of pain and the approval of OxyContin [6]. These factors have resulted in 80 percent of the global opioid supply being consumed in the United States, which represents only 5% of the global population [7]. In 2015 it was estimated that 2.0 million Americans had a diagnosis of opioid use or dependence and in 2018 there were 67,400 opioid overdose deaths [8,9].

Although perioperative exposure to opioids has been clearly identified as a gateway to subsequent opioid abuse, 3–10% of opioid naïve adolescents and adults continue opioid use at 6–12 months following surgery [10,11,12,13,14,15]. The United States leads the world in its post-operative prescribing of opioids, with up to 91% of patients in one study receiving opioid prescriptions after routine surgical procedures, in contrast to only 5% of those patients in the non-USA countries [16]. These data are alarming and must call to action improved efforts to re-evaluate prescriber patterns and take an in-depth look at alternatives to opioids wherever possible. Existing data support opioid-sparing and opioid-free analgesic alternatives (acetaminophen, alpha-2 agonists, regional anesthesia, non-steroidal anti-inflammatory drugs) for many peri- and post-operative procedures. Opioid avoidance is a critical element of all early recovery after surgery (ERAS) guidelines. The perioperative care team, as part of the perioperative home plan, must include clinical and systems-based interventions to alleviate pain with the minimum exposure to opioids [17,18]. This narrative review will present a comprehensive evaluation of the ongoing problem of surgery as a gateway to continued opioid use, and the evidence and options for opioid-free and opioid-sparing alternatives.

The degree of opioid administration in the perioperative period is currently ubiquitous, and it is estimated that up to 99% of patients receive opioids as part of their surgical care [19]. Surgical patients receive opioids at several points during their peri and post-operative course, administered by the surgical and anesthesia team for hemodynamic control and pain relief. Opioids are administered to 84–100% of surgical patients during anesthesia in order to provide a balanced anesthetic and blunt hemodynamic responses to surgical stimuli (hypertension and tachycardia) [19,20,21]. Similarly, in the postoperative period, intravenous and oral opioids continue to be the cornerstone of the management of moderate to severe pain. At discharge in the United States 77–87% of patients having major surgery and 90–91% having minor surgery receive a prescription for opioids [22,23].

The benefits of generous opioid use, however, have been challenged recently with data showing that increased consumption of perioperative opioids is associated with an increase in the incidence of adverse drug events and persistent drug dependence. In the adolescent and young adult population, opioids have been shown to have a particularly high potential for misuse. Studies report that 3.1–10.3% of opioid naïve adolescent and adult patients continue to refill prescriptions [10,11,12,13,14,15]. A large-scale review of a nationwide insurance data set reported that 5.9–6.5% of all patients (21–65 years) demonstrated new persistent opioid use, with no difference in incidence between major and minor surgeries. (Figure 1) New persistent opioid use is defined as the fulfillment of an opioid prescription between 90 and 180 days following the surgical procedure. The degree of surgical pain was not found to be a risk factor for persistent opioid use, indicating that patients continue opioids for reasons other than pain. Similar results were found in analysis of insurance claim data of 88,637 opioid naive adolescents (13–21 years) who underwent surgery. In total, 60.5% of the patients filled a post-operative prescription with narcotics and there was a 4.8% incidence of persistent opioid use in this group [11]. This study also fails to support an association between the magnitude of the surgical trespass (major versus minor surgery) with the number of opioid refills, (Figure 1).

Opioid use disorder has been used to identify the refill of one or more opioid prescriptions 90–365 days after surgery. In 2013, the fifth edition of the Diagnostic and Statistical Manual of Mental Disorders introduced Opioid Use Disorder as a new diagnosis which combined the diagnosis of opioid abuse and opioid dependence [24]. It is defined as a problematic pattern of opioid use leading to clinically significant impairment or distress, as manifested by at least two defined manifestations within a 12-month period. [10,11,13,14] Understanding the etiology of this Disorder is an important step in the diagnosis, treatment and prevention. Persistent post-operative opioid use has not been shown to be a consequence of poorly controlled post-surgical pain. Opiate use and post-operative prescribing patterns has not been shown to differ between major and minor surgeries [10,11,12,13,14,15]. There is no difference in persistent opioid use in postoperative adults and adolescents following major (colectomy, hysterectomy) and minor (wisdom teeth, carpal tunnel, laparoscopic appendectomy, tonsillectomy, inguinal herniorrhaphy) surgical procedures [10,11,12,13,14,15]. The incidence has been reported to range from 3–6.9% [14,15]. The non-surgical counterparts, however, do not report such high incidences: Matched non-surgical control groups have a 0.1% incidence of filling an opioid prescription in the same time period [10,12]. Tobacco and alcohol use, substance abuse, anxiety, mood disorders and preoperative pain disorders are independent risk factors for post-operative opiate abuse but the greatest risk factor and predictor is the initial exposure to opioids in the perioperative period [12].

The magnitude of the causal role of surgical opioid exposure to opioid use disorder in opioid naïve patients is significant. In the United States, there are over 51 million surgeries performed annually [25]. Extrapolating from this data, it is estimated that each year over 2.5 million adolescents and adults will transition to persistent opioid use as a consequence. One million of these patients will develop opioid use disorder following a minor ambulatory procedure [26].

There are other justifiable reasons to avoid opioids in addition to the potential for long-term opioid addiction. Opioids have a multitude of acute side effects including nausea, pruritis, respiratory depression, ileus, and constipation. Opioid-related adverse drug events occur at a rate of 13.6 and 82% for moderate and minor events in surgical patients [19,27,28,29]. Patients with opioid-related adverse drug events have significantly increased treatment costs, lengths of stay, and readmissions to the hospital compared to patients who do not have an opioid related adverse event [19,27,28,29]. This invariably decreased patient satisfaction [19]. Poorly controlled pain is also a major contributor to increased length of stay but all of the determinants of postoperative pain are less than clear. Despite the recognition of the problem of acute and persistent postsurgical pain, and despite the generous and ubiquitous use of opioids, 28–63% of surgical patients continue to experience moderate to severe pain in the first few days following surgery [30,31,32]. While opioids are necessary to treat postoperative pain, it is unclear why, with the abundant use of opioids, pain management has not substantially improved over the past decade. Notably, surgical procedure type is not a key determinant of postoperative pain trajectory [30]. Patient specific factors (female sex, younger age, and scoring high on anxiety) are more highly associated with increased opioid use and a worse pain trajectory than surgical procedure [30,32].

The overprescribing of opioids in the post-operative period is a widespread problem. It is well documented that the prescribing patterns of surgeons vary widely, particularly with respect to opioids intended for children and adults [23,33,34,35,36,37,38,39]. It is estimated that between 42–72% of all post-operative opioid tablets are not consumed [22,35]. A study of 12 common surgical procedures showed that of the (average) 30 hydrocodone pills prescribed post-operatively, only 9 were consumed [37]. The number of pills dispensed for an inguinal herniorrhaphy ranged from 15–120, with 33% of the pills going unused [22]. Studies of this nature across surgical specialties consistently document similar patterns.

Opioids continue to have a legitimate role in the treatment of severe breakthrough surgical pain and better opioid stewardship is the sole responsibility of the medical profession [40,41]. The number of pills a patient receives anchors expectations and is the strongest predictor of the amount of opioids the patient will use [42,43]. Just as increased portion size has contributed to increased food intake fueling the obesity epidemic, increased prescription quantities lead to increased opioid consumption [44]. The risk for persistent opioid use, then, increases dramatically for patients still taking opioids on day 3–5 with an astounding 15% and 30% incidence of persistent opioid use if the initial prescription is for more than 8 days and over 31 days, respectively [43].

There are numerous contributors to the overprescribing of opiates in the perioperative period. Overprescribing is not with malintent, but rather it is out of concern for inadequate post-operative pain control, increased emergency department visits and hospital readmissions, increased phone call burden to office staff, and patient dissatisfaction [42]. This behavior unintentionally leads to excess numbers of opioids in the community available for potential abuse by both the patient and household members and is an identifiable target for modification.

The opiate use problem is not only attributed to surgical prescribing patterns. Intraoperative opioid administration may be implicated in the development of opioid-induced hyperalgesia (OIH), a state of nociceptive sensitization following even brief exposures to opioids that results in increased pain sensitivity and increased recovery opioid requirements [45,46]. In the immediate post-operative period opioids that were given intraoperatively can paradoxically make pain worse by shifting the dose response curve to the left and lowering the pain threshold at baseline, even in the presence of low serum levels of opioids. (This is in contrast to opioid tolerance which is distinguished by a right shift of the opioid response curve with habituation of the opioid receptors as a function of length of exposure.) Surgery is, regrettably, a vulnerable setting for development of OIH. OIH has been demonstrated in both experimental models and human clinical trials [47,48,49,50,51,52]. Abdominal surgery patients receiving higher doses of both fentanyl and remifentanil manifested increased opioid requirements in the PACU, when randomized to higher (vs lower) opioid dosing [47,53]. Remifentanil has been more widely studied and more strongly implicated in OIH but, at least one study has shown that OIH was more pronounced with fentanyl when compared to remifentanil [49,51]. The preoperative, preemptive dosing of oxycontin, a common practice in joint replacement surgery, also may result in increased postoperative pain and opioid requirements [54]. Esmolol infusions administered during surgery decrease intraoperative opioid requirements, and while not classified as an analgesic medication, subsequently decreased opioid requirement in the recovery room [55,56,57]. This is currently attributed to the avoidance of OIH. Anesthesiologists can, therefore, mitigate OIH by avoiding or reducing the delivery and prescribing of opioids in the operative and immediate post-operative recovery period. One institution reports a 1.5 point decrease in the 11 point Numeric Rating Pain Scale (0 is no pain and 10 is maximum pain) for PACU admission pain scores when intraoperative opioid administration was decreased by 30% [17]. The amount of intraoperative opioid used does not, however, correlate (negative or positive) with postoperative pain trajectory, again, underscoring the importance of individual patient risk factors [17,30]. Literature is emerging on opioid-free and opioid-sparing alternatives to traditional opioid-based pain management for both intraoperative and postoperative applications [41,58,59,60,61,62]. Many readily available drugs and analgesic techniques can reduce or eliminate the use of opioids before, during, and after surgery [42,58,59,60,62,63]. There is a continued movement for national and international societies to advocate for a multimodal (multidrug) approach to analgesia with several available published guidelines [64,65,66]. Multimodal strategies, such as the American Society of Anesthesiologists Perioperative Pain Management Guidelines, are designed to activate the receptors of different distinct pain pathways, capitalizing on different modes of action to additively or synergistically control pain. Nonopioid analgesics that are currently available and demonstrate opioid-sparing properties include acetaminophen, nonsteroidal anti-inflammatory drugs (ketorolac, ibuprofen, celecoxib), alpha-2 agonists (dexmedetomidine, clonidine), N-methyl-d-aspartate (NMDA) receptor antagonists (ketamine, dextromethorphan), lidocaine, glucocorticoids (dexamethasone), Nefopam hydrochloride, (a centrally acting analgesic that inhibits the reuptake of serotonin, norepinephrine and dopamine) and magnesium [64,67,68,69,70,71,72,73,74]. Meta-analyses demonstrate consistent opioid sparing ranging from 24–100% when these medications are used individually and in combination [67,68,69,70,71,72]. Thirty-nine RCTs utilizing low-dose intravenous ketamine for postoperative analgesia after a variety of surgeries concluded that ketamine provides a 40% reduction in overall opioid administration [75]. The optimal specific combination of non-opioid analgesic agents and techniques will depend on surgery (e.g., site of surgery, surgical approach) and patient (e.g., presence of comorbidities or medical contraindications) factors. Opioid-free and opioid-sparing analgesia not only minimize the adverse effects of opioids outlined above but enable earlier ambulation and food intake [76]. Modifying the individual dosing of different medications decreases the incidence of their attributable adverse events.

Multimodal analgesic strategies are not limited to enteral and parenteral medications and an abundance of literature documents the excellent opioid-sparing properties of regional anesthesia (RA) [62,63,65]. RA, the temporary blockade of nerves to provide numbness to an isolated area of the body, is used to render a surgical area insensate. A variety of RA techniques block nerves at the spinal cord (spinals and epidurals) or target peripheral nerves (upper extremity {brachial plexus}, lower extremity {femoral and sciatic}, and truncal blocks {ilioinguinal, paravertebral}). RA, regardless of nerve location, uniformly minimizes opioid consumption and provides superior postoperative analgesia and fewer opioid-related adverse events when compared with opioid analgesia across a broad range of minor and major surgeries in children and adults [45,77,78,79,80,81,82]. The use of RA has facilitated same day surgical discharge for a large number of painful orthopedic procedures that were previously performed on an inpatient basis. Studies in children and adults demonstrate a 60–100% reduction in hospital opioid use and a 60–100% decrease in unanticipated hospital admissions [79,80]. Opioid consumption on days 2–3 were unchanged due to the single injection nature of the nerve blocks (duration limited by the pharmacokinetics of the local anesthetic) but the use of in-hospital and home catheters can extend the ability to provide analgesia for several days through the most painful phases of recovery with added opioid sparing [78]. A Cochrane Library review (45 RCTs) of femoral nerve blocks for acute postoperative pain after knee replacement found consistent opioid sparing with regional anesthesia across all studies as high as 73% and 68% at 24 and 48 h, respectively [83,84,85,86,87]. A comprehensive review of regional anesthesia is beyond the scope of this paper and the optimal choice of regional anesthetic(s) will depend on surgical and patient factors. Unique barriers to the wider acceptance of regional anesthesia exist. There is a learning curve of healthcare personnel involved in the performance and management of regional anesthesia. Anesthesiologists must have additional medical knowledge and procedural skills, ward personnel need to monitor the patients and provide additional therapy accordingly, hospital administrators need to support capital equipment purchases (ultrasound machines) and surgeons and patients need to understand the rationale and associated benefits. Surgeons tend to lack widespread enthusiasm and view RA as unnecessary and time-consuming and disruptive to the workflow of the operating room [88]. Patient misperceptions about regional anesthesia are also common and include an exaggerated fear of potential complications resulting in patient refusal [89].

The goal of opioid-free and opioid-sparing anesthesia and analgesia, however, is not without challenges [20,58,90,91,92]. Several institutions have published their models and experience and have identified challenges and solutions specific to decreasing opioid use and increasing non-opioid alternatives. Challenges included reluctance of team members to participate in standardized protocols and operating room and recovery room nurse apprehension about eliminating intraoperative opioids [58,90]. Physicians in particular need to demonstrate an increased willingness to follow guidelines [92]. Pilots in the airline industry adhere to strict guidelines and checklists and are considered exemplary in improving safety. It would be unacceptable to fly with a pilot refusing to use the checklist. Solutions included identifying a committed multidisciplinary team to identify mitigatable opportunities, set goals, develop well-defined plans (guidelines/protocols), facilitate communication and education, and establish a supportive culture that emphasizes a willingness to learn and change [58,90]. It goes without saying that ideal team members need an eagerness to succeed, and resources need to be provided for multimodal opioid alternatives.

## 2. Conclusions

In the United States, approximately 5% of opioid naïve surgery patients develop long term opioid dependence with the single most predictive factor being exposure during the surgical experience. Physicians treating surgical pain then unintentionally contribute to the opioid crisis through current opioid prescribing and administration patterns, limited use of multi-modal therapy and failing to identify risk factors for long term opioid use disorder. Many causal elements are, however, identified and amenable to mitigation with coordinated efforts towards opioid stewardship with the implementation of opioid-sparing and opioid free strategies. Surgical patients are ideal candidates for carefully designed care pathways that limit opioid exposure throughout the perioperative period. It is the responsibility of all members of the surgical team, especially anesthesiologists and surgeons, to acknowledge the problem, develop processes and provide resources to overcome barriers.

## Figures and Tables

**Figure 1 jcm-10-01472-f001:**
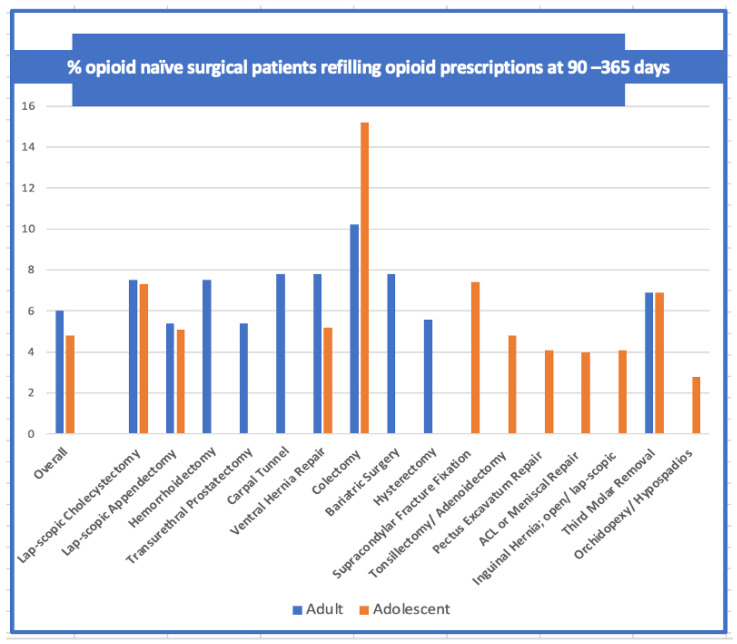
The incidence (%) of patients who filled an opioid prescription after surgery with prolonged opioid refills (*y*-axis) among opioid-naïve patients undergoing surgery by procedure (*x*-axis) are shown. The incidence of persistent use following major (hysterectomy, colectomy, etc) is not statistically different following minor surgery (carpal tunnel, third molar extraction, etc.) [11,12,15].

## Data Availability

Not applicable.

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
