# Peer review of "In the Arms of Morpheus without Morphia; Mitigating the United States Opioid Epidemic by Decreasing the Surgical Use of Opioids"

_jcm, 2021, doi:10.3390/jcm10071472_

Round 1
Reviewer 1 Report
I think the authors provided an engaging narrative review of the problem of perioperative opioid use fueling the opioid epidemic in the United States. while there are a number of similar reviews as well as ongoing research in this field, overall, i think the article was clearly written, provided a reasonably up to date review of the topic and provided motivation to find actionable interventions clinicians could employ. with regards to previous related review articles, according to national library of medicine pubmed.gov, there were 253 articles referencing 'perioperative opioid use review' in 2020 alone.
Perhaps what makes this article unique is the rather deep dive the authors take on evidence of perioperative opioid use and its postoperative ramifications in terms of ongoing opioid use for even years after exposure. however, there are some surprising dissenting data in this regard that the authors did not include (see Vasilopoulous T, et al. Patient and procedural determinants of postoperative pain trajectories. Anesthesiology. 2021;134(3):424-434. in this prospective observational cohort study, intraoperative opioid administration was not found to be associated with postoperative pain trajectory).
The authors spend a few lines on means to reduce intraoperative opioid use. I would recommend removal of references to gabapentoids (pg5 ln176) as per the findings published Verret M, et al. Perioperative use of gabapentinoids for the management of postoperative acute pain: a systematic review and meta-analysis. Anesthesiology. 2020;133(2):265-279. On the other hand, there is no reference in this section to the use of esmolol infusion to reduce postoperative pain. please consider adding.
There are minor spacing and punctuation errors scattered about the manuscript that should be improved. for example:
- pg1 ln35 there is a space btw 10 and % that shouldn't be there
- pg2 ln 47, there is a large space btw the end of the sentence and the references
- pg2 ln 54, there is a large space btw to 84-100%
- generally throughout the ms sometimes the authors put reference numbers immediately after the period ending the sentence, sometimes they don't
- please check the % on pg2 ln64
- pg 2 ln72, correct "adolescents( 13-21 years)"
- all three figures have inadequate captions, literally just reference from the original source. the figures literally maintain the original caption and then the authors' caption sites the original source sort of. please do better with the captions. the authors did have permission to essentially reprint these original figures? figure 3 needs more context.
Author Response
Thank you for your time to review and provide suggests to improve the manuscript. They were very helpful and I hope have added to the value of the publication.
Perhaps what makes this article unique is the rather deep dive the authors take on evidence of perioperative opioid use and its postoperative ramifications in terms of ongoing opioid use for even years after exposure. however, there are some surprising dissenting data in this regard that the authors did not include (see Vasilopoulous T, et al. Patient and procedural determinants of postoperative pain trajectories. Anesthesiology. 2021;134(3):424-434. in this prospective observational cohort study, intraoperative opioid administration was not found to be associated with postoperative pain trajectory).
Thank you for pointing this article out. The content is applicable to this manuscript and has been incorporated in two different places:
Poorly controlled pain is also a major contributor to increased length of stay but all of the determinants of postoperative pain are less than clear. Despite the recognition of the problem of acute and persistent postsurgical pain, and despite the generous and ubiquitous use of opioids, 28-63 % of surgical patients continue to experience moderate to severe pain in the first few days following surgery.24–26 While opioids are necessary to treat postoperative pain, it is unclear why, with the abundant use of opioids, pain management hasn’t substantially improved over the past decade. Notably, surgical procedure type is not a key determinant of postoperative pain trajectory.24 Patient specific factors (female sex, younger age, and scoring high on anxiety) are more highly associated with increased opioid use and a worse pain trajectory than surgical procedure.24,26 and:
The amount of intraoperative opioid used does not, however, correlate (negative or positive) with postoperative pain trajectory, again, underscoring the importance of individual patient risk factors.12,24
The authors spend a few lines on means to reduce intraoperative opioid use. I would recommend removal of references to gabapentoids (pg5 ln176) as per the findings published Verret M, et al. Perioperative use of gabapentinoids for the management of postoperative acute pain: a systematic review and meta-analysis. Anesthesiology. 2020;133(2):265-279. On the other hand, there is no reference in this section to the use of esmolol infusion to reduce postoperative pain. please consider adding.
Reference to gabapentinoids have been deleted and esmolol added:
Esmolol infusions administered during surgery decrease intraoperative opioid requirements, and while not classified as an analgesic medication, subsequently decreased opioid requirement in the recovery room.49–51 This is currently attributed to the avoidance of OIH.
There are minor spacing and punctuation errors scattered about the manuscript that should be improved. for example:
- pg1 ln35 there is a space btw 10 and % that shouldn't be there
- pg2 ln 47, there is a large space btw the end of the sentence and the references
- pg2 ln 54, there is a large space btw to 84-100%
- generally throughout the ms sometimes the authors put reference numbers immediately after the period ending the sentence, sometimes they don't
- please check the % on pg2 ln64
- pg 2 ln72, correct "adolescents( 13-21 years)"
All minor errors have been corrected.
- all three figures have inadequate captions, literally just reference from the original source. the figures literally maintain the original caption and then the authors' caption sites the original source sort of. please do better with the captions. the authors did have permission to essentially reprint these original figures? figure 3 needs more context.
I was waiting to obtain the copyright permissions but decided instead to replace the photocopied figures with one original to this manuscript that is more integrated and allows direct comparison of pediatric and adult data for same surgeries. Captions and legends are complete.
There are other scattered minor spacing and punctuation errors that should be easily corrected.
I tried to find them all.
Reviewer 2 Report
Thank you for submitting the manuscript. I have read your paper very carefully. The topic is certainly interesting and well developed. I suggest some small changes which, in my opinion, could improve the quality of the manuscript.
1)In the introduction you write:
“His uncle was, fit- 30 tingly, the god of death and morphia or morphine is named after him. Thus, begins the history of the opioid epidemic.” Although the reference to classical culture and the myth of Morpheus is suggestive, in my opinion this passage is risky. In fact, the anesthetist, because of his prerogatives should be compared to the Greek God Hypnos (Yπνος), father of Morpheus.
Morpheus does not specifically have the task of inducing sleep, but that of shaping dreams, so much so that the Greek root from which this name derives (μορφή) means form, for which Morpheus is the one who assumes the form. Apart from this mythological digression it seems to me really risky to say that the opioid epidemic originates from Morpheus. There are over two thousand years apart, as well as enormous cultural differences. Please rephrase this passage and consult this site: www.theoi.com
2)In lines 107 to 114 you deal with the possible side effects of opioid use. This reflection is certainly worthwhile, also in terms of costs. However, I suggest comparing the effects, even important ones, of the use of opioids, to those of incorrect post-operative pain management. Pain is in fact one of the main factors that influence the length of hospitalization, and there is evidence that acute pain not adequately treated can turn into chronic pain. In my opinion, the positive effects of opioids, such as cardiac preconditioning in cardiac patients, should also be highlighted in the discussion.
3)In lines 214 to 224 you write about the challenges of opioid-free anesthesia. In my opinion, some topics should be mentioned: the learning curve of healthcare personnel involved in the management of regional anesthesia (anesthesiologists who must perform it, ward personnel who must monitor the patient and administer the therapy), the means not always available in every center (eg ultrasound) and sometimes the patient's refusal of regional anesthesia.
4)Finally, I ask you to delete the quote at the end of the paper (“Society still looks to the medical profession for help and for hope during difficult times. This is one of those times”), replacing it replacing it with a less strong one.
5) Finally, since the paper is focused on the US situation, while in some countries of the world we have the opposite problem, that is an under-use of opioids, I ask you to rethink the title, immediately leading the reader to understand what the focus of the paper is.
I have read and reviewed your paper with great pleasure, which is excellently written, with timely and updated bibliographic references. I hope my comments have been helpful to you.
Kind Regards
Author Response
First, let me thank you for your time and effort to review the paper. Your comments are very helpful and I think have resulted in an overall improved, more informative manuscript. Please find specific changes below:
Please rephrase this passage and consult this site: www.theoi.com
rewritten:
Morpheus is the Greek god of dreams and when in his embrace, an individual enjoys a deep, peaceful sleep; referred to as "being in the arms of morpheus." Morpheus scattered the seeds of poppies over weary mortals. Morphia or morphine is thus named after him. Hence begins the story of morphine.
2)In lines 107 to 114 you deal with the possible side effects of opioid use. This reflection is certainly worthwhile, also in terms of costs. However, I suggest comparing the effects, even important ones, of the use of opioids, to those of incorrect post-operative pain management. Pain is in fact one of the main factors that influence the length of hospitalization, and there is evidence that acute pain not adequately treated can turn into chronic pain. In my opinion, the positive effects of opioids, such as cardiac preconditioning in cardiac patients, should also be highlighted in the discussion.
Excellent points; Revised as follows:
Poorly controlled pain is also a major contributor to increased length of stay but all of the determinants of postoperative pain are less than clear. Despite the recognition of the problem of acute and persistent postsurgical pain, and despite the generous and ubiquitous use of opioids, 28-63 % of surgical patients continue to experience moderate to severe pain in the first few days following surgery.24–26 While opioids are necessary to treat postoperative pain, it is unclear why, with the abundant use of opioids, pain management hasn’t substantially improved over the past decade. Notably, surgical procedure type is not a key determinant of postoperative pain trajectory.24 Patient specific factors (female sex, younger age, and scoring high on anxiety) are more highly associated with increased opioid use and a worse pain trajectory than surgical procedure.24,26
3)In lines 214 to 224 you write about the challenges of opioid-free anesthesia. In my opinion, some topics should be mentioned: the learning curve of healthcare personnel involved in the management of regional anesthesia (anesthesiologists who must perform it, ward personnel who must monitor the patient and administer the therapy), the means not always available in every center (eg ultrasound) and sometimes the patient's refusal of regional anesthesia.
Revised to include:
There is a learning curve of healthcare personnel involved in the performance and management of regional anesthesia. Anesthesiologists must have additional medical knowledge and procedural skills, ward personnel need to monitor the patients and provide additional therapy accordingly, hospital administrators need to support capital equipment purchases (ultrasound machines) and surgeons and patients need to understand the rationale and associated benefits.
4)Finally, I ask you to delete the quote at the end of the paper (“Society still looks to the medical profession for help and for hope during difficult times. This is one of those times”), replacing it replacing it with a less strong one.
Done.
5) Finally, since the paper is focused on the US situation, while in some countries of the world we have the opposite problem, that is an under-use of opioids, I ask you to rethink the title, immediately leading the reader to understand what the focus of the paper is.
Changed to:
In the Arms of Morpheus Without Morphia; Mitigating the United States Opioid Epidemic by Decreasing the Surgical Use of Opioids.
I have read and reviewed your paper with great pleasure, which is excellently written, with timely and updated bibliographic references. I hope my comments have been helpful to you.
Thank you very much.